# An Integrative Analysis of Nasopharyngeal Carcinoma Genomes Unraveled Unique Processes Driving a Viral-Positive Cancer

**DOI:** 10.3390/cancers15041243

**Published:** 2023-02-15

**Authors:** Xiaodong Liu, Yanjin Li, Xiang Zhou, Sinan Zhu, Neslihan A. Kaya, Yun Shen Chan, Liang Ma, Miao Xu, Weiwei Zhai

**Affiliations:** 1School of Life Sciences, Division of Life Sciences and Medicine, University of Science and Technology of China, Hefei 230027, China; 2Key Laboratory of Zoological Systematics and Evolution, Institute of Zoology, Chinese Academy of Sciences, Beijing 100101, China; 3Department of Medical Oncology, Sun Yat-sen University Cancer Center, State Key Laboratory of Oncology in South China, Collaborative Innovation Center for Cancer Medicine, Guangzhou 510060, China; 4Centre for Quantitative Medicine, Health Services & Systems Research, Duke-NUS Medical School, Singapore 169857, Singapore; 5Genome Institute of Singapore, Agency for Science, Technology and Research, Singapore 138672, Singapore; 6Bioland Laboratory—Guangzhou Regenerative Medicine and Health Guangdong Laboratory, Guangzhou 510005, China; 7Center for Excellence in Animal Evolution and Genetics, Chinese Academy of Sciences, Kunming 650223, China

**Keywords:** NPC, integrative analysis, driver genes, copy number variations, mutational signatures, genome instability

## Abstract

**Simple Summary:**

Nasopharyngeal carcinoma (NPC) is extremely frequent in several regions of the world, particularly Southern China (known as the Cantonese cancer). Previous studies were often limited to small cohorts, and systematic comparison with other cancer types has not been explored. In this study, we collected the largest NPC cohort (*n* = 431) and provided the first integrative analysis of NPC genomes. We identified many novel drivers and mutational signatures in NPC. By comparing NPC with other cancer types, we discovered many unique processes driving a viral-positive cancer type.

**Abstract:**

As one of few viral-positive cancers, nasopharyngeal carcinoma (NPC) is extremely rare across the world but very frequent in several regions of the world, including Southern China (known as the Cantonese cancer). Even though several genomic studies have been conducted for NPC, their sample sizes are relatively small and systematic comparison with other cancer types has not been explored. In this study, we collected four-hundred-thirty-one samples from six previous studies and provided the first integrative analysis of NPC genomes. Combining several statistical methods for detecting driver genes, we identified 25 novel drivers for NPC, including *ATG14* and NLRC5. Many of these novel drivers are enriched in several important pathways, such as autophagy and immunity. By comparing NPC with many other cancer types, we found NPC is a unique cancer type in which a high proportion of patients (45.2%) do not have any known driver mutations (termed as “missing driver events”) but have a preponderance of deletion events, including chromosome 3p deletion. Through signature analysis, we identified many known and novel signatures, including single-base signatures (*n* = 12), double-base signatures (*n* = 1), indel signatures (*n* = 9) and copy number signatures (*n* = 8). Many of these new signatures are involved in DNA repair and have unknown etiology and genome instability, implying an unprecedented dynamic mutational process possibly driven by complex interactions between viral and host genomes. By combining clinical, molecular and intra-tumor heterogeneity features, we constructed the first integrative survival model for NPC, providing a strong basis for patient prognosis and stratification. Taken together, we have performed one of the first integrative analyses of NPC genomes and brought unique genomic insights into tumorigenesis of a viral-driven cancer.

## 1. Introduction

As a rare form of head and neck cancer, nasopharyngeal carcinoma (NPC) has an incidence of <1 per 100,000 per year in most parts of the world, but its incidence can be >100-fold higher in South China (especially in Cantonese), Southeast Asia, North Africa as well as arctic regions (e.g., in Inuit) [1]. Many risk factors, including host genetics [2], Epstein–Barr virus (EBV) infection [3] and smoking and dietary preference (e.g., salted fish) [4], are often associated with incidence of NPC. Understanding the genetic basis of NPC and complex interactions with these risk factors can provide an important basis for prevention and treatment of NPC.

As a rare form of cancer, NPC was not included in The Cancer Genome Atlas (TCGA) and International Cancer Genome Consortium (ICGC), but its genomic landscape has been characterized by several studies from individual research groups [5,6,7,8,9,10]. In general, NPC is a tumor type with relatively low mutation burden driven by mutational processes, including DNA repair and APOBEC signatures [11,12]. Functionally important pathways, such as NF-κB, chromatin remodeling and ERBB-PI3K, are frequently mutated in NPC [8,11,13]. Despite rapid progress in the field, several important gaps remain to be filled. First, the cohort sizes for genomic studies are relatively small (often *n* < 100). For example, despite a handful of significant driver genes, there are a large proportion of patients with no driver mutations (30–60% depending on the study) [5], indicating insufficient statistical power in identifying driver genes in earlier studies. Second, the genomic and statistical tools employed to analyze NPC genomes are becoming outdated and the original findings (e.g., mutational signatures) are becoming incompatible with the latest developments (e.g., COSMIC 3.0) [14]. Last, as a rare cancer type outside large consortiums, the genomic landscape of NPC genomes has not been systematically compared with other cancer types. As one of very few viral-positive cancers, pan cancer comparison can yield unique genomic insights into the tumorigenesis of NPC.

In this study, we first curated from the literature six whole-exome sequencing datasets [5,6,7,8,9,10]. By integrating all the exomes (*n* = 431), we reprocessed the raw sequencing data through a unified pipeline and systematically dissected the genomic landscape of NPC. Using state-of-the-art methods, we characterized driver genes, copy number alterations and mutational signatures in NPC. Using multiple genomic features extracted from different genomic layers, including tumor heterogeneity metrics, we constructed an integrative survival model for NPC. We have performed one of the first integrative analyses of NPC genomes and established an important foundation for precision medicine in NPC.

## 2. Materials and Methods

### 2.1. Data Integration

We collected tumor-normal paired samples’ raw FASTQ sequencing data from six previous studies: (1) Sequence Read Archive (SRA) database (accession No. SRP035573), which included 56 paired nasopharyngeal carcinoma samples [7]; (2) SRA database (accession No. SRA288429), which included 51 paired samples [6]; (3) dbGAP-NHGRI (Study ID: 20055, Nasopharynx Cancer Whole-Exome Sequencing), which included 102 paired samples [8]; (4) dataset from Sun Yat-Sen University Cancer Center, Guangzhou, China [10], which included 59 paired samples; (5) dataset from Sun Yat-Sen University Cancer Center, Guangzhou, China [9], which included 75 paired samples; (6) SRA database of NCBI under accession number PRJNA716262 [5], which included 88 paired samples.

### 2.2. Whole-Exome Sequencing (WES) Data Analysis

With the raw FASTQ dataset, we first cut adapter and filtered low-quality reads using fastp (V0.21.0) [15] and then we aligned reads to human genome build 38 (Hg38) using Burrows–Wheeler Alignment tool (BWA, v0.7.17) [16], Genome Analysis ToolKit (GATK, v4.1.3.0) [17] was used for marking duplicates and base quality recalibration. We used Mutect2 [18] to call somatic mutations and filtered somatic mutations using GATK. Funcotator from GATK was used to annotate mutations. Somatic copy number variation and tumor purity were inferred using Sequenza (v3.0.0) [19].

We further filtered low-quality data based on: (1) the mean sequencing depth of one paired normal and tumor should be both no less than 20; (2) the tumor mutation burden (TMB) should not be lower than 0.3 per Mb; (3) to avoid tumor-normal mismatch, the proportion of somatic mutations in dbSNP database should be less than 0.8. After filtering, 363 paired samples were maintained from the 431 samples for further analysis.

### 2.3. Identification of Driver Genes and Timing of Somatic Mutations

To identify driver genes, we used five different methods: MutsigCV (v1.1.4) [20], dNdScv (v0.0.1) [21], CBaSE (v1.1) [22], CHASMplus (v1.0.0) [23] and MutPanning (v2.0) [24]. We used different filtering criteria for different methods. For MutdigCV, dNdScv, CBaSE and MutPanning, we extracted genes with adjusted *p*-value less than 0.1. For CHASMplus, we retained the genes with mean CHASMplus score > 0.6 and corrected *p*-value less than 0.01. In saturation analysis, we only used MutsigCV to find driver genes.

We calculated cancer cell fraction (CCF) of all mutations in the tumor according to purity and copy number using the method provided in the R package EstimateClonality (v1.0) [25]. Mutant allele frequencies and copy number alterations can be obtained by mutect2 and sequenza, respectively. We subsequently divided the mutation into early and late mutations based on the binomial test implemented in the package to determine whether the mutation of a gene is more likely to occur at an early or late stage of tumorigenesis.

GISTIC2 algorithm was used to calculate significantly mutated CNVs at arm and focal levels [26]. To examine the effect of different sample sizes on the significance of CNVs, we randomly selected 100 samples to find significant CNVs with GISTIC2 and compared them with the results from the entire dataset. In order to test co-occurrence and mutual exclusiveness of driver events, we used presence-and-absence information of different genes/pathways, and Fisher exact test was employed to identify significant relationship between different pairs of genes/pathways.

### 2.4. Signature Analysis

SigProfiler [27] was used to discover mutational signatures. First, SigProfilerMatrixGenerator [28] (v1.2.1) was used to create a mutation matrix for single nucleotide variations, double nucleotide variations and INDELs; only variants in exome regions were considered. Then, we used SigProfilerExtractor (v1.1.4) to perform signature detection. De novo signatures were first discovered using an unsupervised method and were subsequently decomposed into COSMIC signatures (v3.3) [14,27,29]. A de novo signature with poor similarity to known signatures will be categorized as a new signature.

### 2.5. Clonal Deconvolution

We used PyClone (v0.13.1) [30] to perform clonal deconvolution. Shannon index (*SI*) [31] was calculated based on clonal structure inferred from PyClone. $SI=-\sum_{i=1}^{n}p_{i}ln\;p_{i}$, where *n* denotes total number of clones and $\;p_{i}$ denotes proportion of clone *i*. *MATH* score [32] was calculated based on the original study, $MATH=\frac{MAD\left(VAF\right)}{Median\left(VAF\right)}\times 100$. Proportion of late-stage mutations (pLM) was calculated based on proportion of late mutations inferred from EstimateClonality [25].

### 2.6. Integrate Survival Analysis

A list of 21 features, including clinical features, ITH features and molecular features, were first curated. We selected driver genes presented in over 15 patients in the subset of patients with survival information (*n* = 232). We filtered copy number variations presented in at least 60% of patients. To describe the relationship between different variables and patient survival, we used a univariate Cox model to calculate hazard ratio and significance (i.e., *p*-value). To find out the importance of different factors, we used a multivariable Cox model, and the proportion of the Wald statistic for each feature was used to represent the importance of each variable. We used the cph function in the rms R package to fit the Cox models.

To test the correlation between different variables, we used different methods for different types of paired samples. For two categorical variables, we used the Fisher test. For two continuous variables, we used Student’s *t*-test. For one categorical variable and one continuous variable, we used Kruskal–Wallis test.

The c index was calculated to represent the prediction accuracy of different features. To calculate the c index, 80% of the samples were randomly selected and used as the training dataset and the remaining 20% samples were used as the test dataset, and 100 repeats were applied to calculate the c index value.

## 3. Results

### 3.1. Compiling the Largest Cohort of NPC Genomes

Even though several studies have been conducted to survey the genomic landscape of NPC [5,6,7,8,9,10], their sample sizes are relatively small. In order to integrate all available NPC genomes, we collected raw sequencing data from the six largest exome sequencing studies from Guangzhou (*n* = 75 and 59, respectively) [9,10], Macau (*n* = 88) [5], Hong Kong (*n* = 51) [6], Singapore (*n* = 56) [7] as well as a combined patient cohort from Hong Kong and Pennsylvania (*n* = 102) [8] (Table 1, Appendix A). In total, 431 paired tumor-normal samples were collected for the integrative analysis. By processing the raw sequencing data using the same analysis pipeline and subsequently filtering away low-quality samples (e.g., low-purity samples, Methods), we maintained 363 high-quality NPC exomes for subsequent integrative analysis.

Using Mutect2 [18], we called somatic mutations across all 363 patients, yielding 64,003 somatic nucleotide variants (SNVs) with 23,787 missense mutations, 2703 insertions, 2463 deletions, 1729 nonsense mutations and 47 pre-stop mutations (Appendix A). The median TMB for NPC is found to be 1.80 per megabase, an intermediate value compared to many other cancer types in the TCGA cohort (Appendix A).

### 3.2. Integrative Analysis Uncovered Many Novel Driver Genes for NPC

Even though there have been many studies investigating driver genes for NPC [5,6,7,8,9,10], statistical power of these studies is often limited (due to small sample sizes), and only frequently mutated genes were often reported. In order to take advantage of the large sample size in this study, we integrated multiple statistical methods for identifying driver genes. In particular, we employed MutsigCV [20] (based on somatic mutation frequencies), dNdScv [21] (based on ratio of non-synonymous to synonymous mutations), CBaSE [22] (based on Bayesian estimation of mutation rates for different mutation types), CHASMplus [23] (a machine learning method taking into account cancer-specific mutational effect) and MutPanning [24] (considering nucleotide context around driver mutations). In total, 38 driver genes were found combining all five methods (Figure 1, Methods). Classical driver genes including *TP53* (11.3%), *CYLD* (8.8%) and *NFKBIA* (8%) are the most frequent driver genes in NPC. In order to understand the novelty of these driver genes, we first curated from the literature a list of known driver genes. From four major studies reporting driver genes with explicit statistical evidence (Appendix A), we curated a list of 145 candidate driver genes (Figure 2a). These driver genes are distributed in several known pathways for NPC, such as cell cycle and growth, NF-κB, chromatin remodeling and organization, PI3K-Akt signaling pathway, autophagy as well as immune-related pathways. In addition to literature reported NPC drivers, we also overlapped the list of driver genes with the Cancer Gene Census (CGC), which is a pan cancer driver gene list curated by the research community [33].

Out of the 38 genes, 13 genes, including classical driver genes, such as *TP53* and *NFKBIA,* overlap with the reported driver list (Figure 2a and Appendix A). These 13 genes included many important drivers, such as *BAP1* (5%, in cell cycle and growth pathway), *TRAF3* (5.8%, NF-kappa B signaling pathway), *ARID1A* (3%, Chromatin remodeling and organization), *FBXW7* (4.1%, autophagy pathway) and *PIK3CA* (3.3%, PI3K-Akt signaling pathway). In addition to the reported driver genes, 12 driver genes not previously reported in NPC were indicated as driver genes in the CGC. They included important driver genes such as *KMT2D* (8.5%), *TGFBR2* (2.5%), *SMAD4* (0.6%) and *STK11* (1.4%). It is worth pointing out that, even though many of these drivers can be found within frequent mutated gene lists, their statistical evidence was lacking previously. It is interesting to observe that all 12 drivers except *ARHGAP5* (2.2%) belong to known pathways for NPC (Appendix A), and we noticed that autophagy- and immune-related pathways are particularly enriched for novel drivers, suggesting that larger cohort sizes from integrative analysis empowered higher statistical power in detecting rare drivers in known pathways for NPC.

In addition to known driver genes (*n* = 13) and novel drivers within CGC (*n* = 12), there are 13 new drivers not reported before for NPC or other cancer types (Figure 2a). For example, *ATG14* (2.5%) plays important roles in promoting membrane tethering and fusion of autophagosomes and endolysosomes [34], and downregulation of *ATG14* can make it sensitized to cisplatin-induced apoptosis in ovarian cancer [35]. *TRAF2* (2.2%) is also considered as a proto-oncogene because of its ability to activate NF-κB pathway in epithelial cancers [36]. It is noteworthy that a large proportion of these driver genes (6/13) are not in known pathways for NPC and warrant future study (Appendix A). Using the ratio of non-synonymous to synonymous mutations (*d_N_*/*d_S_* value) for missense and truncating mutations, we found that many of the driver genes, including *CBFB* and *ATG14*, are potential tumor suppressor genes with high truncating mutations (Figure 2b). Interestingly, when we tested co-occurrence and mutual exclusiveness between driver mutations, we found that, even though there are many pairs of co-occurring drivers with nominal significance, including *TP53* and *KMT2D* (Figure 2c), surprisingly, there are no pairs of drivers with a mutually exclusive relationship, and this is also true at the pathway level (Figure 2c inset). These observations suggest that driver events tend to act concertedly driving tumorigenesis of NPC. One striking observation we found is that NF-κB pathway, the most frequently mutated pathway in NPC, tends to act more solely compared to other pathways, suggesting strength of NF-κB pathways for tumorigenesis of NPC.

In order to understand timing of these driver events, we estimated clonality of driver mutations and assigned clonal mutations as early events and subclonal mutations as late events (Methods). Interestingly we found that mutations in *TP53*, *BAP1*, *FGFR3* and *TGFBR2* tend to be early, while mutations in *C2orf78* and *ITGA10* tend to be late (Figure 1, Methods). When we analyzed the timing of mutations at the pathway level, we found that drivers in cell cycle and growth pathways tend to be very early, while novel genes not belonging to pathways (i.e., others) as well as chromatin organization pathways tend to be late (Figure 2d). These observations suggest that there are still many rare driver mutations driving tumor progression (i.e., being late) that remain to be discovered for NPC.

### 3.3. “Missing Driver Events” in NPC

With a large number of NPC genomes collected in our study, we wondered whether the increased statistical power would enable identifying most of the driver genes for NPC [37]. By downsampling the total dataset to different subsets, we re-performed the driver identification analysis and found that there is a linear relationship between sample size and number of identified driver genes (Figure 2e). Breaking down the driver genes by their frequencies, we found that common driver genes with a frequency 5% or higher have all been discovered and reach saturation at sample sizes around 250. For rarer driver genes, driver genes are still rapidly increasing and are far from saturation (Figure 2e). Future studies with larger sample sizes will likely identify more rare drivers for NPC.

Even though the large sample size endowed us with higher statistical power in detecting driver genes, we still observed a large proportion of patients (164/363, 45.2%) with no driver mutations (Figure 1), a phenomenon we termed “missing driver events (MDE)”. One may hypothesize that MDE might be because the number of driver genes identified in NPC is rather low. When plotting number of identified driver genes as a function of sample size across all cancer types, we found that number of driver genes identified in NPC is comparable to other cancer types (Appendix A). Therefore, it is not the number of driver genes that is responsible for the MDE in NPC. In addition to the number of driver genes, low mutation burden might lead to lower driver events in NPC. When we compared the proportion of tumors without driver genes across cancer types by controlling for sample size (i.e., downsample the TCGA cohort to the same number of patients, Methods), we found that NPC had a typical number of driver genes identified at its corresponding level of TMB (Figure 2f). However, the proportion of tumors without driver genes is significantly higher in NPC (Figure 2g). Taken together, we found that NPC is a cancer type with strong “missing driver events”, possibly driven by unique evolutionary origin (e.g., EBV infection; see Discussion).

### 3.4. Larger Sample Size Empowers Copy Number Identification

In addition to mutational events, copy number variations are the other major molecular events driving cancer progression. By combining all the samples, we found that deletions in chromosome 3p, 9p, 14q, 16q and amplifications of 3q, 8q, 12p and 12q were the most significant CNV events in NPC (Figure 3a) [5]. Leveraging the large sample size from the combined cohort, we used GISTIC algorithm to identify putative driver CNVs across the NPC genome [26]. The increased sample size allowed us to identify more significant focal events than previous individual studies (Figure 3a,b). For example, compared to a random sampled subset (*n* = 100), the statistical significance is much stronger in the combined cohort and more focal events (e.g., amplifications in 2q35 or deletions in 1p36.21) were found in the combined cohort (Figure 3b).

One striking observation identified in CNV analysis is the high frequency of chromosome 3p deletion (80.4%). As NPC genomes tend to have “missing driver events” in point mutations, we wondered whether tumorigenesis in NPC might be compensated by having more frequent CNV events. By combining NPC genomes with all the tumor types from the TCGA, NPC genomes do have more deletions than amplifications, even though the overall amount of CNV events in NPC is not very high compared to other cancer types (Figure 3c). When breaking down frequencies of individual arm events across the TCGA cohort, we found that deletions in chromosome 3p and 14q are indeed extremely frequent in NPC compared to arm levels events observed across cancer types in the TCGA cohort (Figure 3d). In sum, integrative analysis across multiple cohorts empowers copy number identification and revealed a unique landscape of genome instability in NPC.

### 3.5. Novel Mutational Signatures across the NPC Cohort

As a rare form of cancer driven by viral infection, NPC has several unique mutational signatures. In addition to canonical age-related signatures (e.g., SBS1 and SBS5), APOBEC- (SBS2 and SBS13), DNA-repair- (SBS6/15/20/16) and tobacco-related signatures (SBS4 and SBS29) were previously found in NPC [5]. With advancements in signature analysis, new algorithms and signature database (COSMIC 2.0 to 3.0) have been developed, leading to new signatures and terminologies in the signature profile. More importantly, in addition to signatures defined at the single-base level, mutational signatures including double-base substitutions (DBS), indel (ID) [14] as well as copy number signatures (CNV) [29] have been recently developed but are unknown for NPC. Using the latest computational methods [27] together with the largest NPC cohort, we aimed to fully characterize the mutational process in NPC.

Using sigProfiler [27], we found that the overall mutational profile at the single-base level is dominated by C > T substitution (Figure 4a) [8,11], and sigProfiler first identified five de novo signatures and subsequently projected them to sixteen known mutational signatures in the COSMIC database (v3.3). We found that signatures related to aging (SBS1 and SBS5), APOBEC (SBS2 and SBS13), tobacco (SBS4 and SBS29) and DNA mismatch repair (SBS3 and SBS15) were the leading mutational signatures in NPC [11,12] (Figure 4b, Appendix A). In addition to previously reported signatures, we also identified several new signatures, such as SBS7a (UV exposure), SBS10b (Polymerase epsilon), SBS22 (aristolochic acid exposure) and SBS87 (Thiporuine chemotherapy treatment) (Figure 4b). Even though the prevalence of these signatures is much rarer across NPC patients (Figure 4b, Appendix A), they often contributed appreciable frequencies when present in the sample.

In addition to SBS signatures, we also performed doublet base substitute (DBS) analysis and found that AT > NN and TG > NN substitutions are very frequent in NPC (Appendix A). Signature analysis identified only one de novo signature. Even though several known DBS signatures, such as DBS2 (smoking signature), DBS5 (chemotherapy) and DBS11 (APOBEC), are weakly correlated with the de novo signature, their correlations are not high (i.e., cosine distance < 0.8). This implies that this de novo signature might be a new signature not present in the database (Appendix A). In addition to SBS and DBS signatures, we found NPC somatic indels are dominated by 1bp insertion of cytosine (Figure 4c). The overall landscape of indel signature can be deconvoluted to six known and two new indel signatures, including ID2 (slippage during DNA replication of the template DNA strand), ID4 (unknown etiology), ID6 (defective homologous recombination-based DNA damage repair), ID7 (defective DNA mismatch repair), ID8 (DNA double strand breaks by non-homologous DNA end-joining mechanisms) and ID14 (unknown etiology) (Figure 4d, Appendix A). In addition to the known indel signatures, two other new signatures, ID83B and ID83D, are also presented in 171 and 106 patients (Appendix A). Taken together, we identified many new signatures related to DNA repair and unknown etiology in tumorigenesis of NPC.

From signatures based on point mutations and indels, we observed several important processes related to DNA repair and genome instability. One recent progression in dissecting mutational processes in cancer involves copy number signatures [29]. Copy number signatures describe distribution of CNV sizes and genomic context in which CNVs arise. Several important CNV signatures have been identified across cancer types. Using SigProfilerMatrixGenerator [28], we found that copy number landscape of NPC is dominated by short fragments in the background of diploid genomes (Figure 4e). Based on the overall landscape, we deconvolved copy number landscape into seven known signatures, including CN1 (Diploidy), CN2 (Tetraploidy), CN16 (chromosomal LOH, chromosomal instability and 1x whole genome duplication), CN17 (tandem duplication and HRD) as well as CN18, CN19 and CN20 with unknown etiology (Figure 4f, Appendix A). Interestingly, we also identified a novel copy number signature designated as CNV48B (Appendix A). CNV48B is dominated by short genomic fragments in diploid state and is highly prevalent in genomes with high CNV segments (Figure 4f, Appendix A).

Through systematic analysis of mutational signatures across SBS, DBS, indel as well as copy number levels, we have identified many previously known and several novel signatures. Many of the signatures converge to similar biological processes, such as DNA repair, APOBEC and genome instability. In order to systematically explore the relationship among these signatures, we plotted the correlation network between multiple signatures and found that APOBEC and DNA repair are two major processes driving positive correlation between multiple signatures (Figure 4g,h). We noticed that several indel signatures are correlated with CNV signatures, including the two novel indel signatures (ID83B and ID83D), even though CNV signatures in general had weak correlations with other signatures. Interestingly, ID83B is correlated with SBS29 (tobacco chewing), suggesting that ID83B might be related to tobacco exposure. In addition to positive correlations, most of the negative correlations are within signature types (e.g., within SBS). Interestingly, the novel CNV signature CNV48B does not co-occur with any other mutational processes but is negatively correlated with CN1 (Diploidy). Taken together, we unraveled a dynamic landscape of mutational processes driving the tumorigenesis of NPC.

### 3.6. An Integrative Survival Model for NPC

Through several genomic analyses for NPC, several prognostic factors have been identified. For example, patients with NFKB mutations are known to have poor survival [13]. However, no systematic studies have been conducted to construct an integrative survival model for NPC. In addition, since intra-tumor heterogeneity (ITH) has increasingly been recognized as an important factor driving patient clinical outcomes but has not been systematically explored for NPC, we constructed three ITH metrics: (1) percentage of late mutation (pLM), calculated as a fraction of subclonal mutations in a given sample; (2) mutant-allele tumor heterogeneity (MATH) score, which measures distribution of variant allele frequencies across all sites; (3) Shannon’s index, based on subclonal proportions (Methods).

In order to construct an integrative survival model, we first curated a list of potential prognostic factors (*n* = 21), including clinical features (e.g., stage), molecular features (e.g., driver mutations) and ITH features. The molecular features include relatively common driver genes and copy number variations with appreciable frequencies (Methods). Many of these features provide strong stratifying information (Figure 5a–d). In order to understand the relationship between these features, we first plotted the correlation network between all the features (Figure 5e). We found that many of these variables are correlated with each other, particularly surrounding genome instability (GII) and TMB. Using a multivariate Cox model, we combined all the variables into an integrated survival model and ranked the importance of all the variables (Figure 5f). Surprisingly, clinical stage contributes weakly to the survival model, but several important metrics, such as TMB, NFKBIA mutation and Shannon index, are among the strongest predictors of patient survival, suggesting importance of molecular profiling in NPC (Figure 5f). When we combined variables from clinical, molecular as well as ITH levels, we found that the combined model had much more prognostic power than models based on a single layer of variables (Figure 5g). Taken together, we have constructed one of the first integrative survival models for NPC and provided an important basis for precision medicine in NPC.

## 4. Discussion

Using the largest integrative dataset for NPC (*n* = 431), we have systematically explored the genomic landscape of this rare cancer type. Using several state-of-the-art statistical methods for detecting driver genes, we identified 25 novel drivers for NPC, including *KMT2D* and *ATG14*. A pan cancer comparison across tumor types revealed a “missing driving event” phenomenon where a high proportion of patients (45.2%) are found to have no known driver mutations. Through signature analysis, we systematically characterized mutational signatures across single-base signatures (*n* = 12), double-base signatures (*n* = 1), indel signatures (*n* = 9) and CNV signatures (*n* = 8). In addition to several known mutational processes, including APOBEC and DNA repair, we also identified mutational signatures related to UV, tobacco, aristolochic acid exposure and several novel mutational signatures with unknown etiology, revealing an unprecedent dynamic mutational process driving the tumorigenesis of NPC. By combining clinical, molecular as well as intra-tumor heterogeneity features, we constructed the first integrative survival model for NPC. Taken together, we have performed one of the first integrative analyses of NPC genomes and provided an important foundation for precision medicine in NPC.

Through multi-layer analysis of mutational signatures ranging from single-base, double-base, indel as well as copy number alterations, we discovered a large collection of mutational processes driving tumorigenesis of NPC. Several de novo signatures found in this study are new and have not been found in other cancer types, worth further investigating together with other cancer types. One striking observation is that there are many mutational processes related to DNA repair and genome instability that are strongly associated with NPC. Previous studies have found complex interactions between the EBV and host genome [38]. For example, interactions between viral and host genome can lead to hypermethylation of a multitude of cellular tumor suppressor gene loci and of viral genomes [39,40]. Moreover, EBV can also integrate to a host genome, strongly perturbing the local regulatory landscape of a genome [41]. We hypothesized that these mutational processes might be related to complex interactions between the host and viral genomes, worth investigating with a multi-modal omic dataset.

As a viral-positive cancer, integrative analysis unraveled several unique processes first discovered in NPC. First, “missing driver events” are quite exceptional across cancer types (Figure 1 and Figure 2g). Together with high prevalence of deletion events (e.g., chromosome 3p deletion, Figure 3a) as well as a large collection of novel mutational signatures related to DNA repair and genome instability, we hypothesized that viral infection and oncogenes in viral genomes might provide infected nasal epithelial cells strong “self-propelling” capabilities without accumulating driver mutations. Subsequent complex interactions between viral and host genomes might lead to frequent chromosomal CNV events (e.g., chromosome 3p deletions) and a suite of mutational processes driving initiation and progression of NPC. Thus, these observations from “missing driver events”, frequent chromosomal deletions and several mutational signatures might be the “collective molecular phenotypes” of a viral-positive tumor such as NPC. It will be interesting to compare NPC genomes with other EBV-positive cancer types, understanding the effect of viral infection driving tumorigenesis. Second, in many cancer types, the clinical stage of a tumor is often of great significance for patient prognosis and treatment. Surprisingly, the clinical stage was not so strong for patient prognosis in NPC. As the clinical information available in the combined cohort is rather incomplete and patchy, “losing statistical power” in these variables might not be so robust and is worth validating in a future study. On the contrary, molecular features such as TMB and NFKBIA were found to be more powerful in predicting patient prognosis than the clinical stage, suggesting the importance of molecular characterization of NPC. In addition, previous studies found that EBV infection and immune infiltration can also play important roles in patient prognosis for NPC [10,42]. These observations suggest great potential further improving clinical staging of NPC through molecular characterization of the tumor and its microenvironment in the near future [43,44].

Even though we collected the largest genomic cohort and conducted a comprehensive integrative analysis of NPC genomes, there are still several limitations worth improving in future studies. First, the clinical data analyzed in this dataset are rather preliminary (Appendix A). As a viral-positive cancer, both phenotypes associated with the cancer and virus can be informative in patient prognosis [3,45]. Collecting comprehensive clinical information can be quite important for survival analysis. Second, even though we identified several important driver genes, their functional roles are still unknown. For example, even though ATG14 was found to have high levels of truncating mutations (Figure 2b), suggesting its potential role as a tumor suppressor, knocking-down experiments found that suppressing ATG14 does not always promote tumor growth [46]. In addition, genes such as NLRC5 can have complex functions in many processes, including cell migration [47]. Finally, further comparison with other cancer types, in particular other viral-positive cancer types, can yield unique insights into tumorigenesis of NPC. For example, even though NPC was found to have a relatively low mutation burden, its level of heterogeneity was found to be rather high (Appendix A). Understanding the complex interplay between virus and cancer will be an interesting topic in the future.

## 5. Conclusions

Using one of the largest cohorts for NPC, we have identified many novel drivers, mutational signatures unique to NPC. By combining features across clinical, molecular as well as intra-tumor heterogeneity levels, we constructed one of the first integrative survival models for NPC. By comparing NPC genomes with other cancer types in the TCGA, we identified many genomic processes unique to NPC. Taken together, integrative analysis of NPC genomes discovered many unique phenomena in a viral-positive tumor and unraveled unprecedently dynamic processes driving the evolution of NPC.

## Figures and Tables

**Figure 1 cancers-15-01243-f001:**
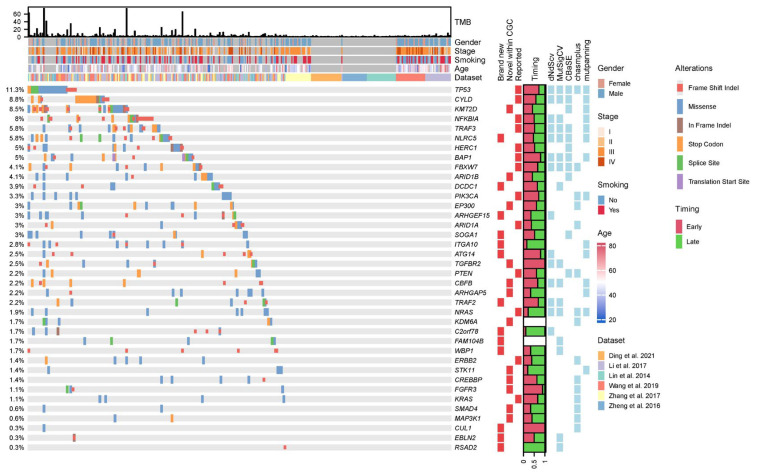
Driver landscape of the NPC cohort. Oncoprint plot of the driver landscape across the patients. The top panel shows the tumor mutation burden (TMB) of the patients. The patient cohort was plotted below the TMB panel. Driver frequencies are shown on the (**left**). Drivers’ status (whether it is reported in the literature (“reported”) or newly identified but within CGC (“novel within CGC”) or brand new), proportion of time being early or late mutations and statistical methods with significant results as driver genes were plotted on the (**right**) [5,6,7,8,9,10].

**Figure 2 cancers-15-01243-f002:**
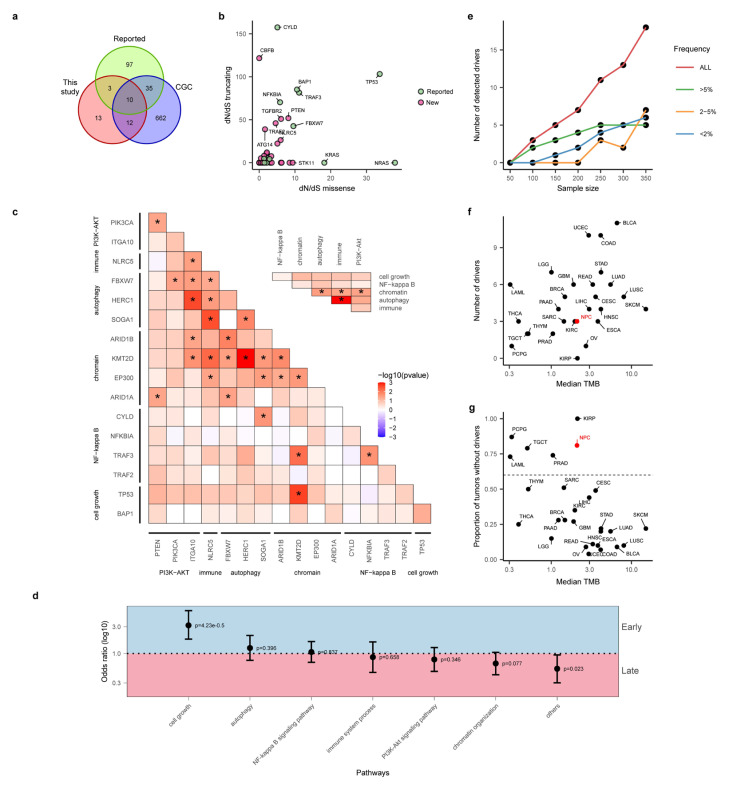
Driver properties and pan cancer comparison. (**a**) Venn diagram between drivers identified in this study (“this study”) from the literature (“reported”) as well as in Cancer Gene Census (CGC). (**b**) *d_N_/d_S_* values based on missense (x-axis) and truncating (y-axis) mutations. Reported drivers are those drivers overlapping between our study and reported driver genes (**a**). (**c**) Correlation between different driver genes (red indicates co-occurrence, blue indicates mutual exclusiveness and * indicates *p*-value < 0.05); the color corresponds to the strength of the correlation (based on *p*-value). Inset figure plotted the correlation between drivers at the pathway level. (**d**) Timing of driver mutations in different pathways. Y-axis plotted the odds ratios of early/late mutations in different pathway genes (as compared to the genomic background). (**e**) The relationship between sample size and number of identified driver genes (i.e., saturation plot). (**f**) The relationship between median TMB and number of identified drivers across the TCGA cohort. NPC was labelled as the red dot. Different tumor types from the TCGA were downsampled to the same size of 100 patients for a fair comparison (Methods). (**g**) The relationship between median TMB and the proportion of tumors without driver genes across the TCGA cohort. NPC was labelled as the red dot.

**Figure 3 cancers-15-01243-f003:**
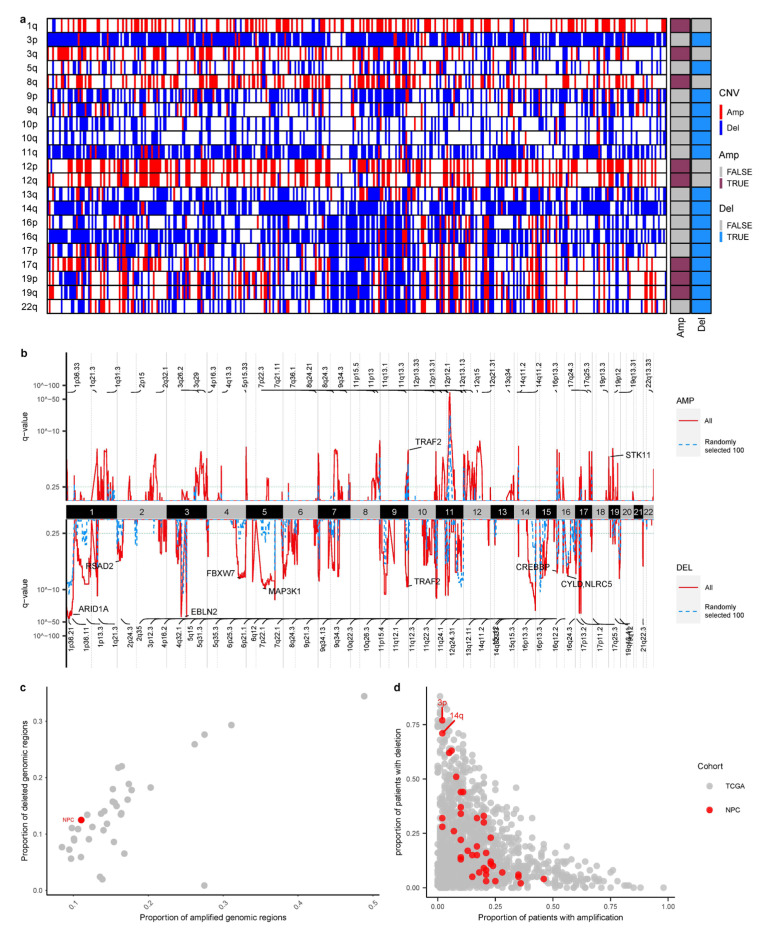
Copy number landscape of the NPC cohort. (**a**) Significant chromosomal CNV changes in the NPC cohort (GISTIC results, Methods). (**b**) Significant focal CNV events identified in the full cohort (all) and randomly subsampled 100 patients (“Randomly selected 100”). Amplifications were plotted on the top; deletions were shown in the bottom. (**c**) Proportion of amplified genomic regions vs. proportion of deleted genomic regions across the TCGA and NPC cohort was plotted. Each dot represents a tumor type. (**d**) For each chromosomal arm, the proportion of patients with amplification (x-axis) vs. proportion of patients with deletion (y-axis) were plotted for all the TCGA and NPC cohorts. Each dot represents a chromosomal arm in a specific tumor type. NPC was shown in red.

**Figure 4 cancers-15-01243-f004:**
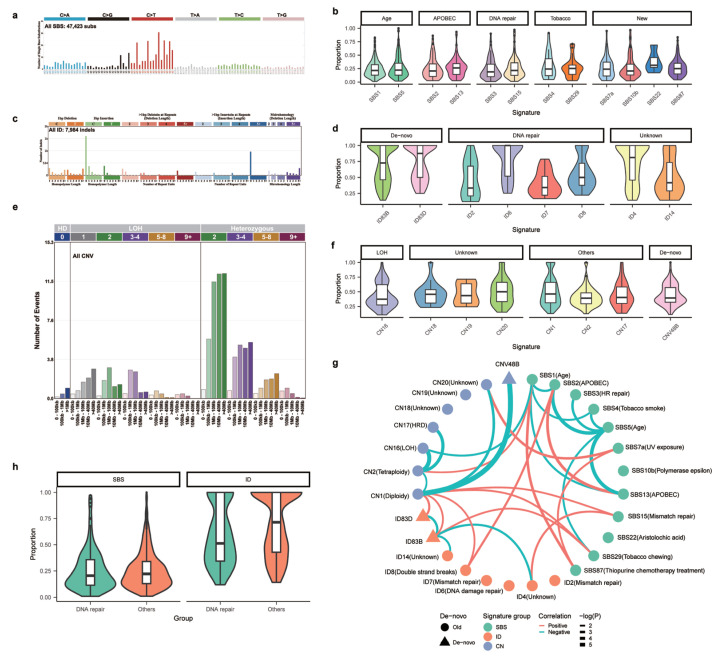
Mutational signatures across the NPC cohort. (**a**) The overall mutational landscape at the single-base level for the NPC cohort. (**b**) Contributions of different SBS signatures in the NPC cohort. (**c**) The overall indel landscape in the NPC cohort. (**d**) Contributions of different indel signatures in the NPC cohort. (**e**) The overall CNV landscape for the NPC cohort. (**f**) Contributions of different CNV signatures in the NPC cohort. (**g**) Correlation plot for different signatures. Red indicates positive correlation and blue indicates negative correlation. Line widths mark the significance of the mutual correlation. (**h**) Contribution of DNA repair signatures vs. other signatures.

**Figure 5 cancers-15-01243-f005:**
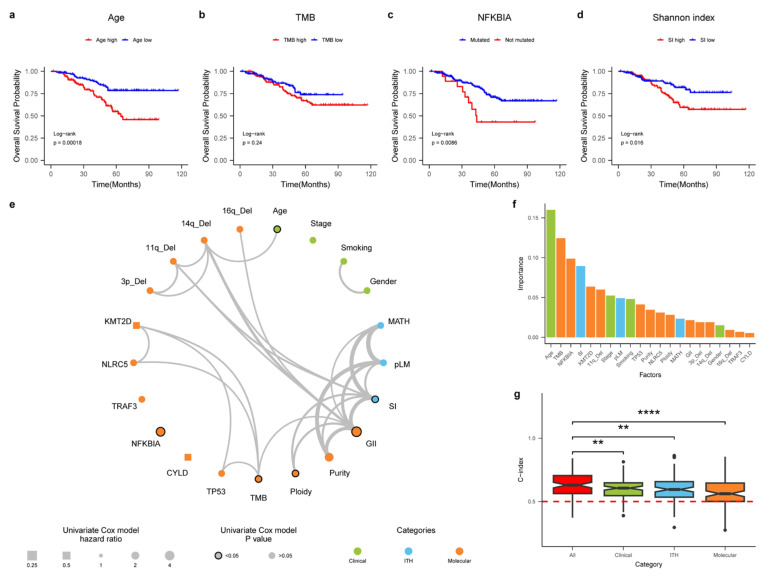
Integrative survival model. (**a**–**d**) The exemplar variables that can stratify patient survival from clinical (age), molecular (TMB), driver (NFKBIA), tumor heterogeneity (ITH). (**e**) The correlation network plot for the features across clinical, molecular as well as tumor heterogeneity (ITH) layers. Line widths marked the significance of the correlation. The dot sizes represented the hazard ratio in the univariate Cox model and margins around the symbols labeled the significance. (**f**) Importance of different features across the clinical, molecular as well as ITH features. (**g**) The predictive accuracy (c-index) of individual models consisting of different features. **/**** labeled statistical differences at *p*-values of 0.01 and 0.0001.

**Table 1 cancers-15-01243-t001:** Clinical table for our cohort.

Group	Percentage (Value)
Observations	
	363
Stage	
I	2.8% (10)
II	8.5% (31)
III	33% (121)
IV	19% (69)
missing	36% (132)
Age	
Mean (SD)	50 (12)
valid (missing)	232 (131)
Gender	
Female	15% (54)
Male	49% (178)
missing	36% (131)
Smoking	
No	28% (103)
Yes	29% (105)
missing	43% (155)

## Data Availability

The data used in this study were retrieved from the original publications.

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
