# Peer review of "An Integrative Analysis of Nasopharyngeal Carcinoma Genomes Unraveled Unique Processes Driving a Viral-Positive Cancer"

_cancers, 2023, doi:10.3390/cancers15041243_

Round 1

Reviewer 1 Report

This is a large integrative genomics analyses of 431 nasopharyngeal carcinoma (NPC) tumours that were collected across multiple studies. The authors leveraged on this pooled cohort to re-run several of the genomics analyses covering driver mutations, copy number aberrations (CNAs), mutational signatures, and attempted to build a prognostic model for NPC. I have the following comments for their consideration.

Comments:

1.   The authors performed a thorough analysis on driver mutations in NPC. Here, they made a few observations: (a) a large proportion of tumours (~47%) with missing driver events (MDEs); (b) a correlation between discovery of driver events and sample size. So, are the authors suggesting that the discovery of driver mutations is yet to be saturated in NPC, based on Figure 2e?

2.   On this note, it would be useful to the readers if the authors indicate what was the criterion used to define a drive mutation?

3.   With regard to the definition of early vs late driver events, how did the authors classify the different mutations? Was it by clonality analysis? If so, did they have all the data to perform such an analysis?

4.   Figure 1: it would also be useful if the authors layered clinical info e.g., TNM stage, age, gender on the Oncoprint.

5.   Figure 2c: this reviewer doesn’t find this result very intuitive. How were the correlation tests performed? Presumably, the heatmap reflects the strength of the association based on the p-values? This should be elaborated in the Figure legend or Main text.

6.   If anything, the analyses on the CNA and mutational signatures are probably the most interesting in the paper. Figure 3b: how were the 100 patients randomly selected? Was it by clinical stratification?

7.   Figure 3d: it would seem that a large proportion of NPC tumours manifest deletions, attributable to 3p and 14q deletion. It would be more interesting to perform another analysis just comparing NPC against other EBV-associated tumours, as opposed to the TCGA cohorts. This would provide some clues if this phenomenon is attributable to EBV infection.

8.   The prevalence of aging mutational signatures in NPC is interesting, since NPC is predominantly a disease of the young (30-50 year-old) Chinese males. Could the authors comment on this finding? Could it be that the median age of their cohort is older, and doesn’t truly reflect the reality for the larger NPC patient population?

9.   This reviewer is particularly intrigued by the CN signatures. The authors ought to elaborate more on the CN signatures since they are newer. Also, from the circus plot in Figure 4g, the CN signatures seem quite exclusive, with only 3 signatures showing an association with other signatures (of which, 2 were aging related signatures). This reviewer is thus more conservative in terms of suggesting associations between the CN signatures and DNA repair/indel signatures.

10.           For the survival analyses, more details must be included, including the clinical parameters (T-category, EBV DNA titre, age, treatment details, etc.), and the derivation of the molecular data, such as ITH, TMB, etc. For example, for ITH, given that multiple methods were used to estimate this, were the different parameters e.g., MATH scores and Shannon indices strongly correlated? This reviewer is also curious if NPC tumours tend to manifest high ITH, especially when they are known to be established following monoclonal expansion of EBV-infected epithelial cells.

11.           Regarding weightage of the different clinical, molecular features, perhaps the authors could utilise Shapley values to demonstrate the weightage of the different features in the model? Finally, perhaps the authors could choose to demonstrate the C-indices for the different models using a conventional AUC plot?

12.           Ultimately, this reviewer would be circumspect with any survival analysis from this pooled cohort, unless the authors had full access to the clinical information from the respective study groups.

Reviewer 2 Report

The authors have undertaken the compilation and re-analysis of six earlier genomic studies of a total of 431 NPC patient genomes. Integrative studies with the most up-to-date bioinformatics tools were undertaken. Utilizing study data from the earlier NGS studies, this larger cohort of compiled genomes allowed the identification of additional rarer driver genes, examined the timing of these somatic mutations, and found new molecular signatures that were not identified in the earlier individual smaller studies.

The integrative analysis revealed a large proportion of NPC tumors do not contain a high tumor mutation load and driver gene mutations. There was a linear relationship between sample size and numbers of identified driver genes. The study confirmed the importance of the NF-kB pathway in NPC tumorigenesis. In this integrative study, 13 new novel driver genes were identified, which have not been reported in other pathways leading to cancer, opening the way for future investigations.

A large number of tumors were missing driver events (MDE), suggesting these tumors were driven by other possible virus-related oncogenesis pathways. Insight into processes driving virus-associated cancers confirmed the importance of copy number alterations resulting from chromosomal deletions and genomic instability, as well as DNA repair pathways contributing to NPC development. The importance of chromosomal 3p and 14q loss and DNA repair genes was verified.

Interesting comparisons to other cancer types was also done. Results validated the importance of the NF-kB pathway and chromatin remodeling, as noted in previous studies, and identified rare driver genes made possible by the larger cohort included in the analysis.

Using new advanced algorithms and updated databases, several new molecular signatures were identified and interestingly implicated UV exposure, Polymerase epsilon, aristolochic acid, and thiporuine chemotherapy.

Important new insight was provided in this manuscript based on results of integrative survival modeling, based on 21 features including clinical, molecular, and intratumor heterogeneity. The group performed the integrative survival modeling, which is useful for predictive patient prognosis. The importance of the tumor mutation burden and the NF-Kb proved to be better predictors of prognosis than clinical staging.

A summary table of the clinical parameters and survival details available for the study should be added to the manuscript. Further discussion on these clinical parameters used to develop predictive prognostic conclusions in their survival model should be presented.

Reviewer 3 Report

Nasopharyngeal Carcinoma (NPC) is rare cancer, so previous genomic studies have a week point that their sample sizes are relatively small. Authors performed integrative analysis of NPC genomes by collecting published genome data. The manuscript is substantially interesting and it may contribute to ameliorate the possibilities of preventing and modulating the biological evolution of NPC in the perspective of a clinical application in humans, and it should be considered for publication.

I would only have some points to improve the article.

Major points

1.    I feel a little confusing about “New and CGC” in Figure 1 and Figure S1. Authors identified 38 driver genes and show “New and CGC” with red squares, but some gene has no square, and some gene has two squares. It may be better to show by three categories, already reported NPC driver genes (n=13), novel NPC drivers within CGC (n=12), and the brand-new driver genes (n=13) which are not reported before for NPC or other cancer types.

2.    Mutations in driver genes can be divided to “loss-of-function” and “gain-of-function”. Can authors assess “loss- or gain-of-function” on the driver genes, especially brand-new driver genes such as ATG14 and NLRC5? If possible, please show the function or discuss about it.

3.    Are there the data about viral infection? Almost NPC in China may be EBV-positive, but how about NPC from Pennsylvania? If there is no data about viral infection, or authors did not compare the genomes between virus-positive and negative NPC, the title should be changed without “a viral positive cancer”.

4.    Authors showed many of new signatures including single base signatures, a double base signature, indel signatures, and copy number signatures. How the molecular characterization of NPC can be applied to patient prognosis and treatment? Can you show any concrete examples?

Minor points

There may be some mistakes, as below.

Line 152: Kruskal–Walli test to be Kruskal–Wallis test

Line 217: … cancer types (Fig. 1b). There is no Fig. b in Figure 1.

Line 288: amplification …, and 18q. In Fig. 3a and 3b, there are no 18q and no amplification in Chr18, respectively.
